# Microstructure, Wear and Corrosion Behaviors of Electrodeposited Ni-Diamond Micro-Composite Coatings

**Xiaoli Wang** [1],*, **Ziyi Zhao** [1], **Jinsong Chen** [2], **Xin Zhou** [1] and **Yinjie Zong** [1]

1. School of Mechanical Engineering, Jiangsu Ocean University, Lianyungang 222005, China
2. Engineering Training Center, Jiangsu Ocean University, Lianyungang 222005, China
* Correspondence: 2006000018@jou.edu.cn

**Abstract:** For the micro-milling of hard and brittle materials, to avoid crack formation, a tool with ductile milling mode is required. Composite electrodeposition technology was used to prepare a Ni–diamond coating on the surface of brass. The surface microstructure, composition and surface roughness of the coating were studied with a scanning electron microscope, X-ray diffractometer and roughness tester. The adhesion strength was studied by scratch test, the wear resistance was analyzed by wear test, and the corrosion resistance was investigated by Tafel curves and electrochemical impedance spectra (EIS). It was found that the distribution of diamond particles of the Ni–diamond coating was relatively uniform, and the content was relatively high. The internal stress of the coating prepared by the composite electrodeposition technology was very low. With the incorporation of the diamond particles, the surface roughness of the coating tended to decrease. The wear experiment showed that the wear scar diameter of the corresponding glass ball for the Ni coating was 1.775 mm and the roughness was $13.88 \pm 2.811$ µm, while that for the Ni–diamond coating was 2.680 mm and $8.35 \pm 0.743$ µm, respectively, indicating that the tool coating with uniform diamond particles had a strong ability to process workpieces with significantly improved surface quality. The particle press-in mechanism not only improved the wear resistance of the coating, but helped to prolong the service life of the tool. The results of the EIS test and Tafel curves showed that the Ni–diamond coating had a lower corrosion current, and the corrosion resistance of the coating surface was improved. The experimental results showed that the micro-diamond coating prepared by the composite electrodeposition technology had good bonding strength, low internal stress, and significantly improved wear resistance and corrosion resistance.

**Keywords:** composite electrodeposition; Ni–diamond; wear; corrosion resistance





## 1. Introduction

Traditional tool materials have poor wear resistance, short life and an easy-to-collapse blade, which cannot meet the processing of hard and brittle materials and easily crack or chip the workpiece [1,2]. Hence, it requires a tool that can produce microfeatures with ductile mode material removal to avoid crack formation and achieve high surface quality; composite coating on the tool is a good solution [3]. Due to the good toughness of brass, it is suitable as the tool substrate during the milling of hard and brittle materials. If the tool surface is properly modified, it can improve the wear resistance of the surface. Inert particles with a Ni-matrix coating were a good solution [4,5]. Diamond is the hardest material in nature, with high hardness, high strength, high wear resistance, small line expansion coefficient, good corrosion resistance and other excellent physical and chemical properties. As an ideal enhanced phase, it is also a special material for the grinding head [6], drill head [7], wire saw [8], grinding wheel [9] and so on, which are widely used in machinery, electronics, construction, drilling, optical glass processing and other industrial fields. Among the preparation methods of Ni-matrix coatings, the electrodeposition technique is commonly used owing to its low operating temperature, high efficiency, low internal stress

and moderate cost [10–12] compared to other coating methods, such as chemical vapor deposition [13,14], physical vapor deposition [15], laser cladding [16] and sputtering [17]. The papers showed that the Ni–diamond coating had good wear resistance [18–21], corrosion resistance [22–24] and was a suitable micro-milling tool for hard and brittle materials.

Huang [18] prepared a high-particle-content Ni–diamond composite coating with a strong abrasive ability using the electrophoretic deposition and electrodeposition techniques. Bao [19] fabricated Ni–diamond composite coatings with burying co-deposition techniques. They found the Vicker's indentation hardness increased with effectively enhanced diamond content in the Ni–diamond coatings, whereas the scratch hardness decreased. Because diamond particles can form lattice distortion and defects, which can significantly enhance the internal stress of the composite coating, the internal stress has a great impact on the scratch performance of the coating. Wang [20] found that the wear resistance of Ni–P–Diamond coatings increased as the diamond particle sizes increased. On the other hand, the larger diamond particles could work as a larger cutter to machine an increased amount of material on the ball and displayed a high wear resistance. Zhang [21] presented a flow chart for the enhancement mechanism of wear resistance for Ni–diamond composite coatings. Modified, the diamond composite condition and the optimal nickel grain size of the composite coating could further increase the microhardness. At the same time, the friction coefficient of the composite coating was decreased by the decreased surface roughness in conjunction with the increased diamond content. The promotion of microhardness combined with the decreased friction coefficient eventually reduced the wear rate of coatings and simultaneously improved the cutting ability, thereby enhancing the overall tribological properties. Zhang [22] and Li [23] proposed an electrochemical co-deposition mechanism. The co-electrodeposition process contains the steps of transfer, weak adsorption, reduction, strong adsorption and continued growth. Wang [25] proposed a micron diamond tool which improved wear resistance, extended the life of the tool, increased yield and reduced cost. The press-in mechanism for diamond particles was originally proposed.

Although many scholars carried out research on Ni–diamond coatings, fewer have applied it as a micro-milling tool surface which is suitable for the processing of hard and brittle materials. In this paper, the surface morphology, composition and surface roughness of the coatings were analyzed with a scanning electron microscope (SEM), X-ray diffractometer (XRD) and roughness tester. The wear resistance was investigated by wear test, and the corrosion resistance was studied by Tafel curves and electrochemical impedance spectra (EIS). By comparing the Ni and the Ni–diamond coatings, it was found the Ni–diamond coating electrodeposited on a good toughness brass substrate not only enhanced surface hardness and wear resistance, but could also easily absorb the vibration in processing. Meanwhile, the binding strength, the wear resistance of the coating and the surface friction characteristics were analyzed and validated through the scratch and wear tests. The press-in mechanism of diamond particles was further studied and provided strong support for improving the wear resistance of micro-milling tool surfaces.

## 2. Experiment

The experiment was divided into two stages, shown in Figure 1. In the first stage, the Cu interlayer was electrodeposited on the brass substrate for 3 min in an electrolyte containing 250 g/L $CuSO_4 \cdot 5H_2O$ and 0.5 M $H_2SO_4$ by applying a current density of 5 A/dm$^2$, as described in our previous works [25,26]. A 200 mL beaker was used as the coating bath. The stirrer was rotated at 100 rpm and the temperature of the solution was kept at 25 °C. The pH value was approximately 0.6. A copper plate of 80 mm × 15 mm × 0.8 mm was used as the anode. Brass sheets of 50 mm × 15 mm × 0.8 mm were used as the cathode, which was sequentially ground by #400, #600, and #800 sandpapers in water, rinsed in deionized water and ultrasonically cleaned in 95 vol% alcohol solution for 5 min. After being dried in air, they were pasted with electroplated insulating tape (Tape Type: NITTO No. 360) to leave an exposed 15 mm × 15 mm area for the deposition. The distance

between the anode and the cathode was 50 mm. Afterward, the sample was flushed with deionized water to remove the electrolyte and temporarily attached materials on the surface. In the second stage, nickel and diamond were compositely electrodeposited on the plated copper [27] in an electrolyte containing 280 g/L $NiSO_4·6H_2O$, 5 g/L $NiCl_2·6H_2O$, 40 g/L $H_3BO_3$ and 5 g/L Saccharin [28–32] with a different current density of 0.67 A/dm$^2$, 1.33A/dm$^2$, 2A/dm$^2$ and 4A/dm$^2$, respectively, as described by Wang [25]. A nickel plate of 50 mm × 25 mm × 0.8 mm was used as the anode and the immersion area was 35 mm × 25 mm. Diamond particles of 2–4 μm were evenly distributed in a porous filter cup (inner diameter: 30 mm, outer diameter: 35 mm, height: 120 mm, aperture: 1 μm; made by Toyo Roshi Kalsha in Japan) with 480 rpm stirring speed, and the concentration was 10 g/L. The brass sample coated with Cu interlayer in the last step was used as the cathode. The overall thickness of the coatings was approximately 8 μm. There were no diamond particles in the solution for the electrodeposited Ni coating, and with diamond particles for the electrodeposited Ni–diamond solution.

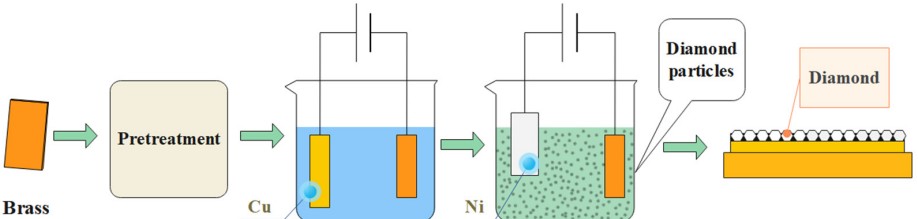

**Figure 1.** Two stages of the composite electrodeposition.

A scratch tester Anton Paar RST$^3$ (ANTON PAAR, Graz, Switzerland) was used to test the adhesion strength of the coating; the scratch length was 5 mm, the scratch speed was 10 mm/min and the initial load was 500 mN, with a final load of 20,000 mN and a loading rate of 39,000 mN/min. The friction and wear tests were carried out on a wear test machine using a ball-on-disk pair under the lubrication of deionized water. The test ball was made of soda–lime silica glass with 6 mm diameter, the Vickers hardness was 481 kgf mm$^{-2}$ and the applied load and the rotating speed were 5.88 N and 90 rpm for 45 min, respectively. The surface roughness of the glass balls was measured with a high-revolution optical microscope (ZETA INSTRUMENTS, San Jose, CA, USA). Eight parallel straight lines were evaluated from the wear scar of the glass ball. The average values and the standard deviations of the roughness along these lines were obtained.

The electrochemical impedance spectrum (EIS) was carried out at the maximum open-circuit voltage (OCP) in a 6.5 g/L solution of potassium hexacyanoferrate (potassium ferricyanide) a CHI 6273 (CH INSTRUMENTS, Austin, TX, USA) electrochemical work-station. The Tafel curves were measured in a 5 wt.% NaCl solution with a voltage setting range of OCP voltage ± 200 mV; the time was 600 s. The OCP measurement time was 1800 s, and each test sample used a replaced solution.

## 3. Results and Discussion

### 3.1. The Comparison of Surface Morphology

Figure 2 shows the surface morphology of the Ni coating and the Ni–diamond coating when the current density was 1.33 A/dm$^2$. Figure 2a shows the surface morphology of the Ni coating, and Figure 2b is a magnified view of Figure 2a. Figure 2c shows the surface morphology of the Ni–diamond coating, and Figure 2d is the magnified view of Figure 2c, with a more uniform and higher content of diamond particles. It can be seen that the diamond particles were tightly surrounded by the nickel matrix and were well bound. Figure 3 shows the distribution of carbon elements of the Ni–diamond coating by EDX at different current densities. It was found that with an increase in current density, the diamond particles in the coating increased and then decreased. When the current density

was 1.33 A/dm$^2$, the content of diamond particles was the highest, and the distribution was the most uniform, the same as the reference [25].

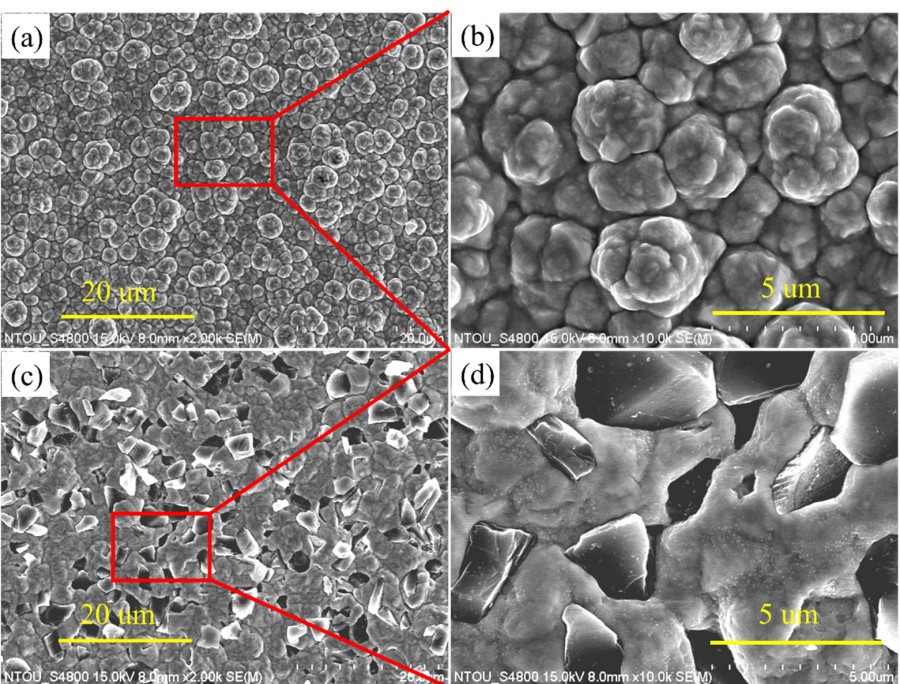

**Figure 2.** The surface morphology by SEM. (**a**) Ni coating, (**b**) the magnified morphology of (**a**), (**c**) Ni–diamond coating, (**d**) the magnified morphology of (**c**).

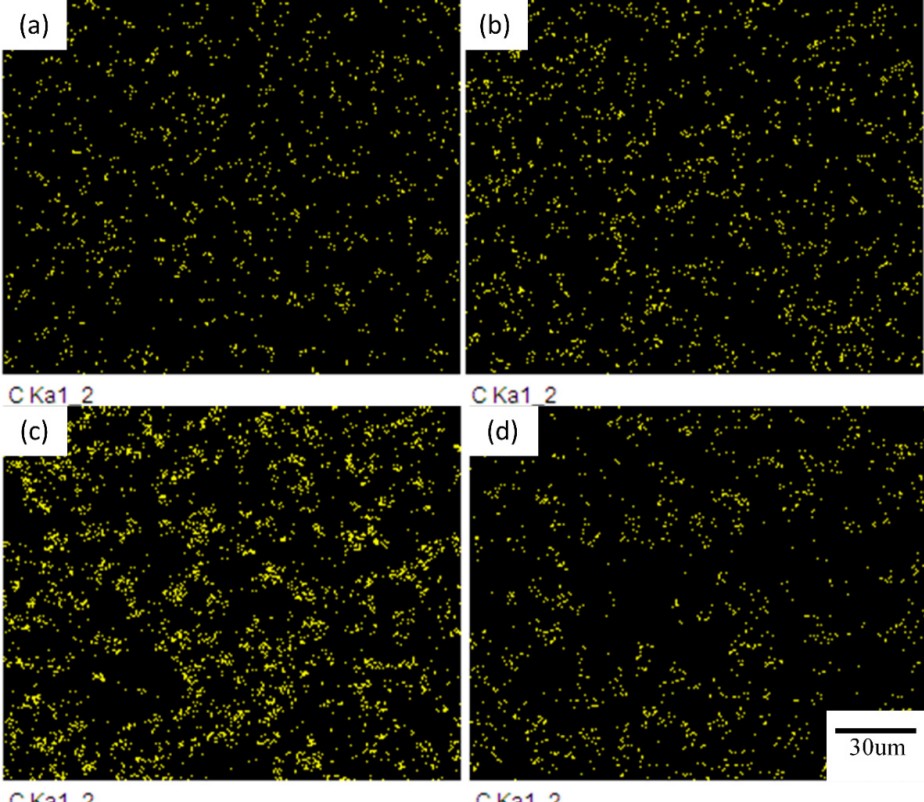

**Figure 3.** The distribution of carbon elements in the Ni–diamond coating by EDX at different current densities. (**a**) 0.67 A/dm$^2$; (**b**) 1.33 A/dm$^2$; (**c**) 2 A/dm$^2$; (**d**) 4 A/dm$^2$.

### 3.2. The Comparison of XRD Spectra

Figure 4a,b show the XRD spectra for the Ni coating and the Ni–diamond coating, respectively. The crystallization of the Ni coating and the Ni–diamond coating were fundamentally the same, with a face-centered cubic (FCC) lattice and consistency with the diffraction standard card of a Ni plate. The crystallization direction was (111), (200) and (220), where (111) was the preferred direction. The peak and crystallization directions of diamond in the Ni–diamond coating corresponded with the JCPDS card number 00-006-0675, the crystallization directions were (111) and (220), the corresponding 2angles were 43.917° and 75.304°, respectively, and they were the cubic crystallization structure. The difference between Figure 4a,b is that (200) of the Ni–diamond coating is significant. The relationship between the X-ray diffraction broadening effect and the crystallite size can be expressed in the Scherrer Equation (1):

$$D = \frac{K\lambda}{\beta \cos\theta} \tag{1}$$

where D is the crystallite size, $\lambda$ = 1.54056 Å is the wavelength of the X-ray, $\beta$ is the full width at half maximum of the dominant diffraction peak at 45°, $\theta$ is the diffraction angle and K is the constant with a value of 0.9.

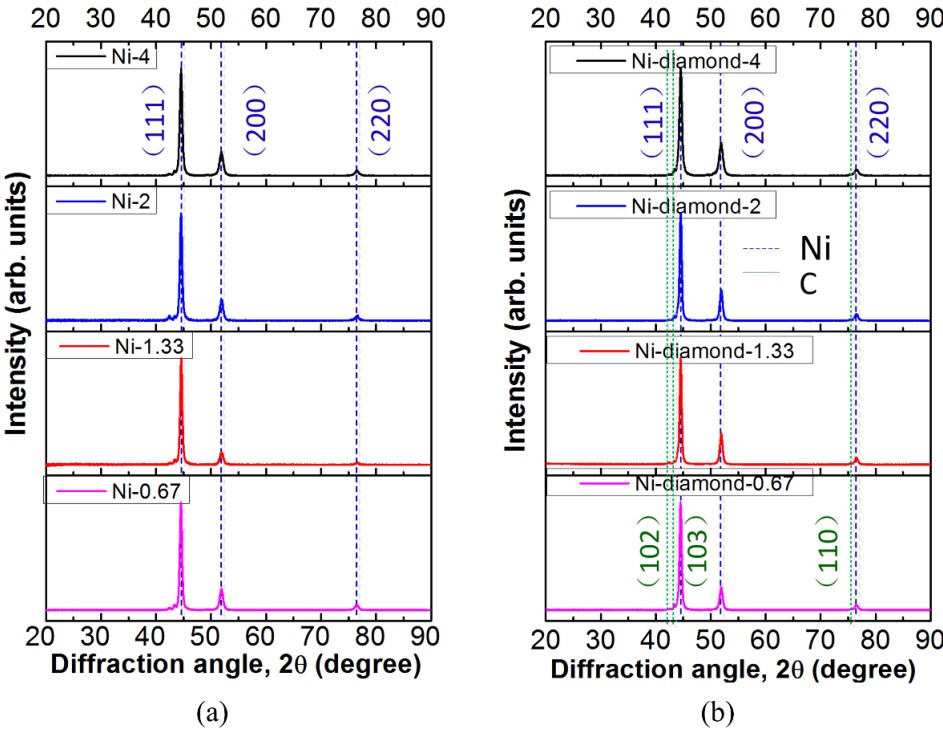

**Figure 4.** XRD spectra. (**a**) Ni coating; (**b**) Ni–diamond coating.

Based on the XRD data, half-height width and 2$\theta$ angle were calculated. The crystallite sizes were calculated using the Scherrer formula, shown in Table 1. As can be seen from Table 1, with an increased current density, the crystallite size of the Ni coating and the Ni–diamond coating tended to decrease, but the current density could not be too large; otherwise, the coating quality deteriorated and had pinhole bubbles. At the same current density, the crystallite sizes of the Ni coating and the Ni–diamond coating were nearly the same. By the comparison of the 2$\theta$ angle of the crystallization direction (111) and Figure 4, it can be concluded that the angle offset of the Ni coating and the Ni–diamond coating was minimal, proving that the internal stress was very low and showing the advantage of composite electrodeposition.

**Table 1.** The comparison of grain sizes of the Ni coating and the Ni–diamond coating at different current densities (0.67 A/dm$^2$, 1.33 A/dm$^2$, 2 A/dm$^2$, 4 A/dm$^2$).

| (<111>) | β (Radian) | 2θ (°) | Crystallite Size D (nm) |
|---|---|---|---|
| Ni | 0.3634 | 44.47 | 23.64 |
| Ni-0.67 | 0.4705 | 44.53 | 18.26 |
| Ni-1.33 | 0.4682 | 44.53 | 18.35 |
| Ni-2 | 0.4988 | 44.53 | 17.22 |
| Ni-4 | 0.5258 | 44.53 | 16.34 |
| Ni–diamond-0.67 | 0.4638 | 44.51 | 18.52 |
| Ni–diamond-1.33 | 0.4458 | 44.51 | 19.27 |
| Ni–diamond-2 | 0.4539 | 44.57 | 18.93 |
| Ni–diamond-4 | 0.5443 | 44.55 | 15.78 |

*3.3. The Comparison of Surface Roughness*

Figure 5a shows the surface roughness of the Ni coating and the Ni–diamond coating. It was found that the roughness of the Ni–diamond coating was greater than that of the Ni coating at the current density of 0.67 A/dm$^2$, but when the current density was 1.33 A/dm$^2$ and 2 A/dm$^2$, the roughness of the Ni–diamond coating was less than that of the Ni coating, which may be due to the diamond particles filling the unevenness of the Ni coating. Therefore, the evenness of the Ni coating could be improved at the suitable current density due to the decreased surface roughness [20] and uniform morphology shown in Figure 3. Figure 5b shows the roughness measurement positions, where 3 point was in the center, and others were 2 mm from the edge. The roughness value is the average of five positions.

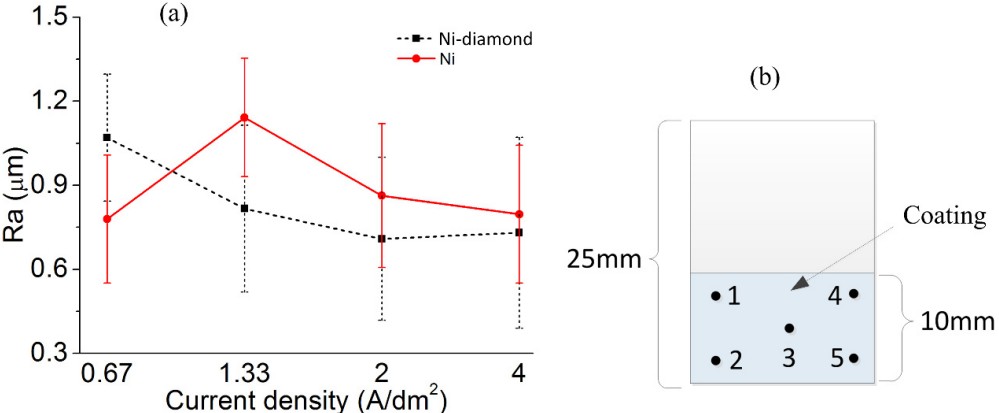

**Figure 5.** The comparison of surface roughness. (**a**) The roughness of Ni coatings and Ni–diamond coatings built by different current densities. (**b**) The sampling positions for surface roughness measurement. Position 3 was located at the center. Positions 1, 2, 4 and 5 were 2 mm from the edges.

*3.4. The Results of Scratch Tests*

Figure 6 shows the end scratch depths for the Ni coating and the Ni–diamond coating. The end scratch depths were 26,864.59 nm and 21,720.15 nm, respectively. By comparison, the scratch depth of the Ni–diamond coating was lower than that of the Ni coating because the surface hardness and scratch resistance of the coating increased with the addition of diamond, limiting the scratch depth. Figure 7a,b show the scratch tracks of the Ni coating and the Ni–diamond coating. Figure 7c,d show the scratch tracks of the Ni coating at the one-third and end positions of the scratch length, respectively. Figure 7e,f show the magnified image of Figure 7c,d, respectively. Figure 7g,h show the scratch tracks of the Ni–diamond coating at the one-third and end positions of the scratch length, respectively, and Figure 7i,j show the magnified image of Figure 7g,j, respectively. It could be seen from Figure 7i that the diamond particles were abundant and distributed uniformly. Although the end of the scratch depth was scratched to the substrate, which was much deeper than

the coating thickness, many diamond particles were still seen in the topography of Figure 7j, indicating that the diamond particles were pressed into the substrate during the scratching process. The press-in mechanism of the particles helped to prolong the service life of the tool as well as improve the wear resistance of the coating. On the other hand, when the coating was scratched through, there was no obvious critical load, which also showed that the bonding of the coating was strong.

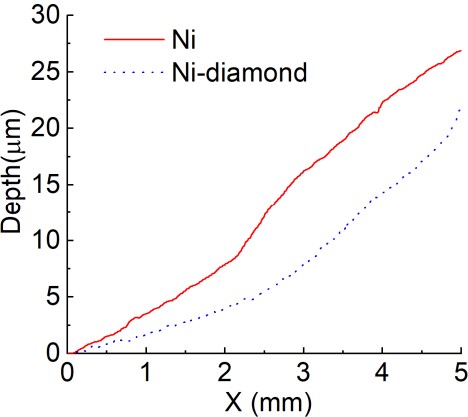

**Figure 6.** The end scratch depths for the Ni coating and the Ni–diamond coating.

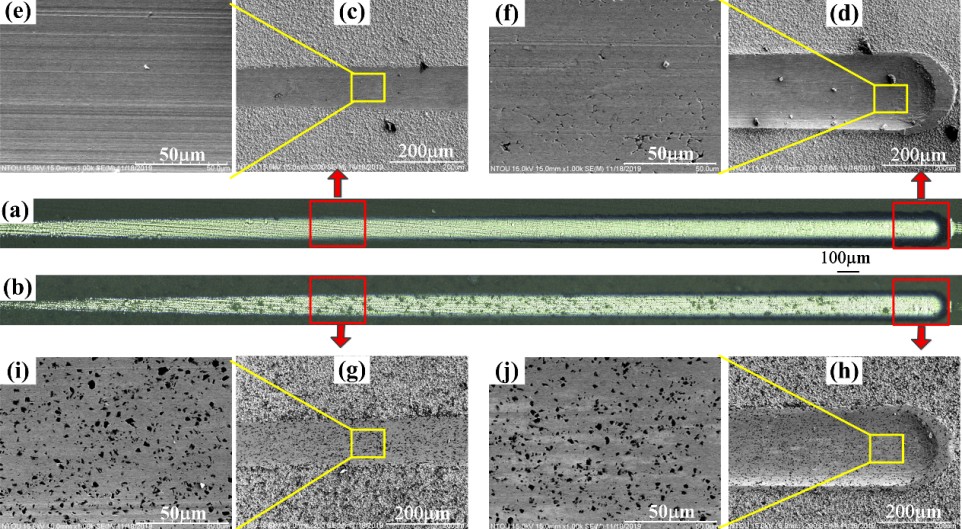

**Figure 7.** The scratch tracks of the Ni–diamond coating and the Ni coating. (**a**,**b**) are scratch tracks of the Ni coating and the Ni–diamond coating, respectively. (**c**,**d**) are scratch tracks of Ni coating at the one-third position and end position, respectively. (**e**,**f**) are magnified scratch tracks of (**c**,**d**). (**g**,**h**) are scratch tracks of the Ni–diamond coating at the one-third position and end position, respectively. (**i**,**j**) are magnified scratch tracks of (**g**,**h**), respectively.

### 3.5. The Results of Wear Tests

The pin-on-disk wear test was conducted, and a glass ball was used as the counterpart. Figure 8 shows the results of the wear test for the Ni coating and the Ni–diamond coating. Figure 8a shows a surface image of the glass ball of the Ni coating; the diameter of the wear track was 1.775 mm, and the surface roughness was 13.88 ± 2.811 μm. Figure 8b shows the magnification of Figure 8a. Figure 8c shows the worn morphology of the Ni coating; it can be seen that the outstanding Ni grain was flattened, but the surface evenness was poor, resulting in the uneven surface of the glass ball. Figure 8f shows the surface image of the glass ball for the Ni–diamond coating; the diameter of the wear track was 2.680 mm. The wear track of Figure 8f is larger than Figure 8a, which may mean that the

diamond particle in the matrix can not only act as a load-bearing phase but also act as micro-cutters. Meanwhile, the existence of diamond particles will reduce the direct contact between the Ni matrix and the ball during the sliding process, which will weaken the adhesion wear between the tribo-pairs and prevent the wear process [19]. Figure 8g shows the magnification of Figure 8f, with uniform and plowed grooves; the width was nearly the same as the diamond particle size, indicating that the surface morphology was the result of diamond milling; the surface roughness was 8.35 ± 0.743 μm. Figure 8h shows the worn morphology of the Ni–diamond coating. After the wear process, there were still many diamond particles on the surface, indicating that the particles were pressed into the coating during the wear test [18]. The press-in mechanism improved the wear resistance of the coating, which was consistent with a previous study [25]. The surface machining quality of the glass ball worn by the Ni–diamond coating was better than that of the Ni coating. Figure 8d,e, as well as Figure 8i,j, show the compositions of the glass balls and the coatings; it was found that the Ni composition was not transferred to the glass ball, but the compositions of the glass ball (Si Ca Mg O, etc.) were transferred to the coating, which indicated that the coating did not contaminate the workpieces and was consistent with a previous paper [25].

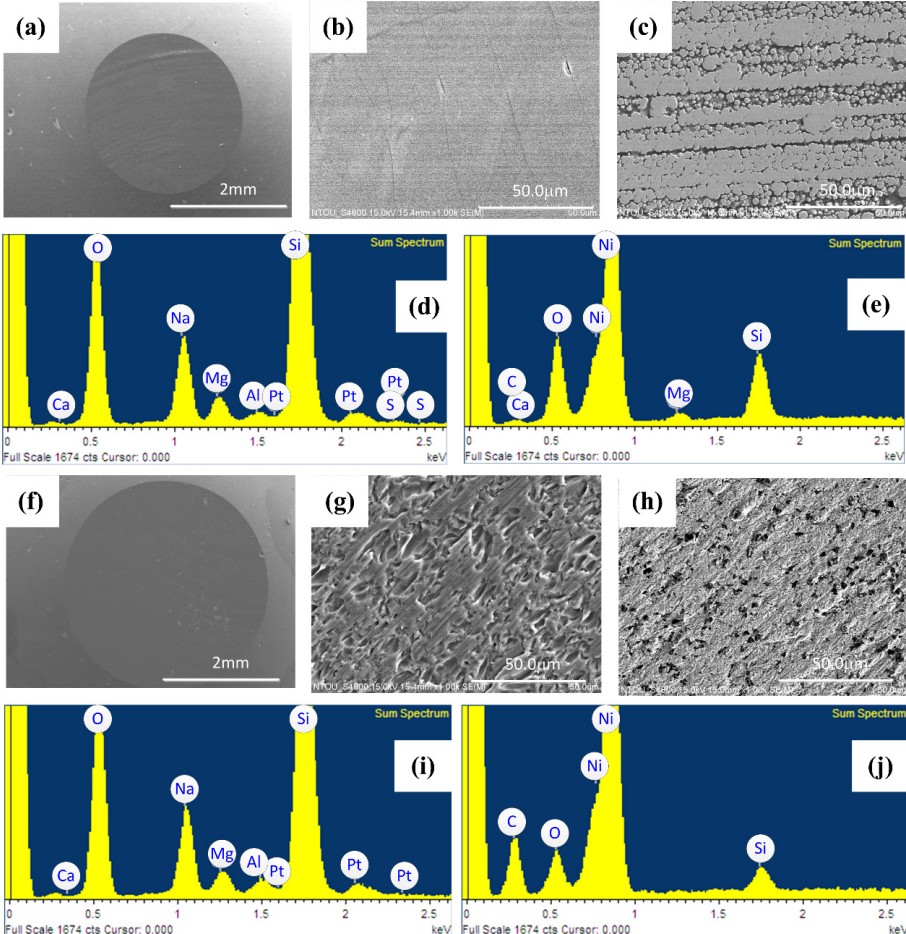

**Figure 8.** Results of the wear test for the Ni coating and the Ni–diamond coating. (**a**) The surface image of the glass ball of Ni coating, (**b**) the magnification of (**a**,**c**) the worn morphology of Ni coating. (**d**,**e**) the compositions of the glass balls and the Ni coatings respectively. (**f**) The surface image of the glass ball of Ni-diamond coating, (**g**) the magnification of (**f**,**h**) the worn morphology of Ni-diamond coating. (**i**,**j**) the compositions of the glass balls and the Ni-diamond coatings respectively. A glass ball was the wear counterpart. The operational parameters of the wear test were 5.88 N and 90 rpm with 4000 cycles.

Figure 9 shows the change in the friction coefficient for the Ni coating and the Ni–diamond coating during the wear tests. It was found that the coefficient of friction of the Ni–diamond coating was lower than that of the Ni coating, which was consistent with Awasthi [33] and Hou [34]; there were two reasons for this. On the one hand, the 2–4 um diamond particles filled the pit on the coating surface, making the surface smoother, and reduced the coefficient of friction. On the other hand, in the wear test, the diamond particles were pressed into the Ni matrix or the brass substrate and reduced the surface coefficient of friction, which was contrary to the results of Huang [18] and Bao [19]. We think that the different diamond sizes and sliding conditions may be the reason for the opposite results because the small diamond size could easily cause particle aggregation and affect the quality of the coating. The friction coefficient was consistent with the surface roughness due to the uniform morphology, so the surface machining quality was better.

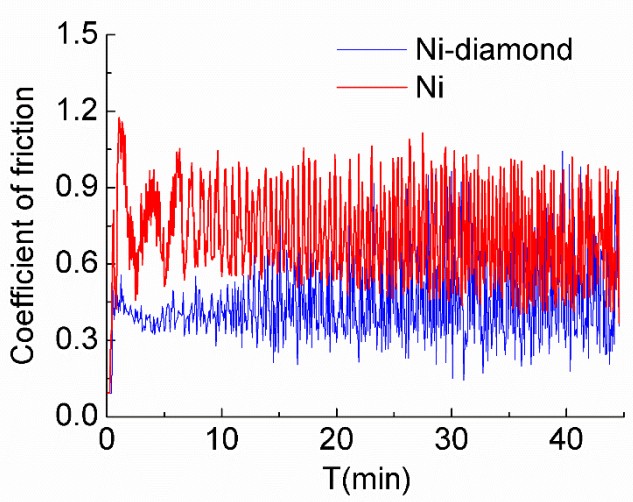

**Figure 9.** The friction coefficient of the Ni coating and the Ni–diamond coating during wear test.

### 3.6. Results of Electrochemistry Test

Figure 10 shows the fitting results of the EIS test and the equivalent circuit for the Ni coating and the Ni–diamond coating. Table 2 shows the impedance value of the equivalent circuits. $C_1$ was the double layer capacitance, and $R_{sol}$ was the solution resistance. $R_1$ was the charge transfer resistance, which could reflect the corrosion resistance. The value of $R_1$ and corrosion rate changed in inverse proportion, and a higher $R_1$ meant better corrosion resistance [22]. The surface impedance of the Ni–diamond coating was slightly increased compared to the Ni coating, which was due to the non-conductivity of the diamond particles. Therefore, the addition of diamond particles has the tendency to increase surface impedance.

Tafel curve was carried out in a 5 wt.% NaCl solution. Figure 11 shows the Tafel curve of the Ni coating and the Ni–diamond coating; it was found that the Ni–diamond coating had a lower corrosion current and better corrosion resistance. There are two factors that can explain an increase in corrosion resistance. First, the incorporated diamond particles play a major role and act as an inert physical barrier to the initiation and development of the corrosion attack [35]. Second, the closely packed (111) crystallization probably contributed to an increase in the corrosion resistance because dense and close-packed crystallization often shows a higher dissolution resistance due to the total energy required for breaking of the bonds, and the subsequent dissolution of atoms is high [35,36].

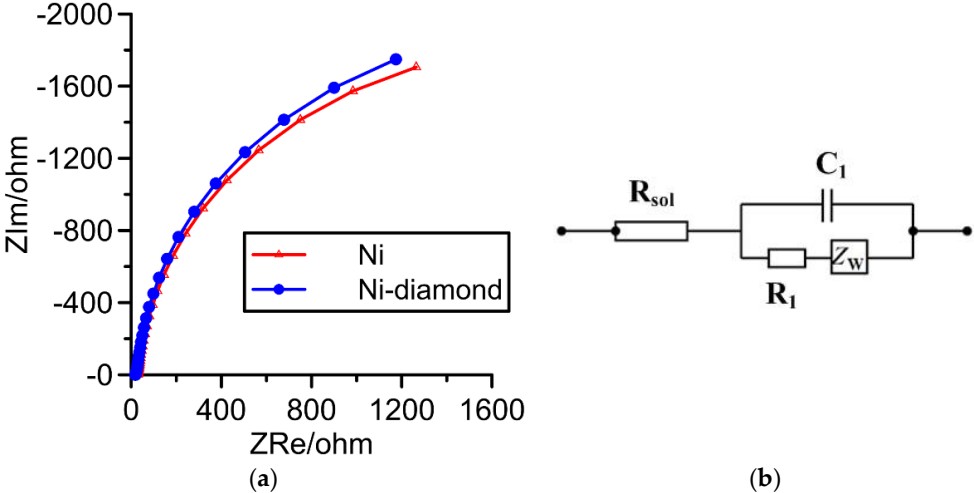

**Figure 10.** The fitting results of EIS tests (**a**) and the equivalent circuit (**b**).

**Table 2.** The impedance value of the equivalent circuits.

| Coating Layers | $R_{sol}$ (ohm) | $R_1$ (ohm) | $C_1$ (F) | $Z_W$ (ohm) |
| --- | --- | --- | --- | --- |
| Cu_Ni | 27.78 | 3728 | $0.6293 \times 10^{-3}$ | 0.01262 |
| Cu_Ni–diamond | 18.44 | 3960 | $0.6510 \times 10^{-3}$ | 0.01357 |

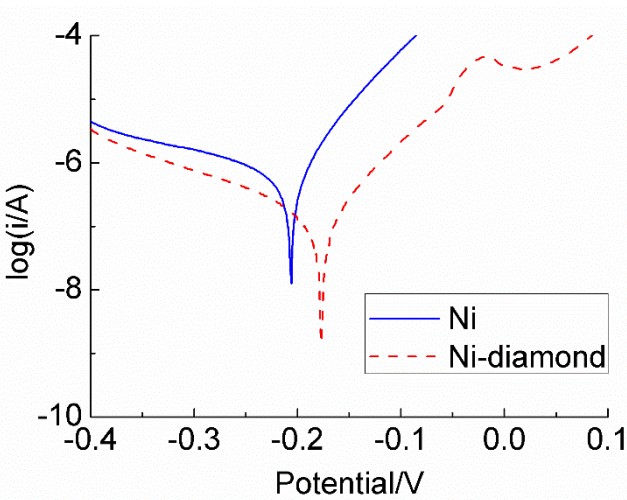

**Figure 11.** The Tafel curves of the Ni coating and the Ni–diamond coating.

## 4. Conclusions

In this study, the Ni–diamond coating was prepared by composite electrodeposition technology, and the surface morphology, composition, surface roughness, wear resistance and corrosion resistance were investigated.

(1) With an increase in current density, the diamond particles and the distribution uniformity increased and then decreased. The angle offset of the crystallization direction of the Ni–diamond coating was minimal, indicating the low internal stress and the advantage of composite electrodeposition. In addition, the Ni–diamond coating had a tendency to be less rough than the Ni coating, which helped to improve surface evenness.

(2) The results of the scratch test and the wear test showed that the Ni–diamond coating had a strong processing capacity and high surface-machining quality. The press-in

mechanism of particles improved the wear resistance and helped to extend the service life of the tool.

(3) The results of the EIS test and Tafel curve showed that the Ni–diamond coating had a lower corrosion current, which demonstrated that the corrosion resistance was enhanced.

**Author Contributions:** Formal analysis, J.C.; investigation, Z.Z. and X.Z.; methodology, X.W.; project administration, X.W.; resources, J.C. and X.Z.; supervision, X.W.; visualization, Y.Z.; writing—original draft, X.W. and Z.Z.; writing—review and editing, X.W. All authors have read and agreed to the published version of the manuscript.

**Funding:** This research was funded by the Graduate Research Innovation Program Project in Jiangsu Province, grant number KYCX21_3141, Haiyan Plan of Lianyungang city, Jiangsu blue and green engineering project, and the Marine Resources Development Institute of Jiangsu in China, grant number JSIMR202022.

**Institutional Review Board Statement:** Not applicable.

**Informed Consent Statement:** Not applicable.

**Data Availability Statement:** Not applicable.

**Conflicts of Interest:** The authors declare that they have no known competing financial interests or personal relationships that could have appeared to influence the work reported in this paper.

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
