# Peer review of "Microstructure, Wear and Corrosion Behaviors of Electrodeposited Ni-Diamond Micro-Composite Coatings"

_coatings, doi:10.3390/coatings12101391_

Round 1
Reviewer 1 Report
The study described in the article and devoted to the analysis of the microstructure and properties of electroplated Ni-diamond coatings is of little originality and is of little interest to the reader. According to the research presented by the author of the article, a lot of work has been done, for example, the following are known: M. Wang, Preparation of Ni-diamond composite coating by composite electroplating. Journal of Central South University (Science and Technology) 44(7):2688-2695. 2013. Xiangzhu He, Preparation and Investigation of Ni-Diamond Composite Coatings by Electrodeposition. Nanoscience and Nanotechnology Letters 4(1):48-52, 2012. E.C. Lee, A study on the mechanism of formation of electrocodeposited Ni–diamond coatings, 2001 and others.
There are a number of comments on the article:
1. Specify clearly at the end of paragraph "1 Introduction", the purpose of the work. Then there are the research questions.
2. In the text of the article, it is necessary to clearly state for what it is necessary to perform this work.
3. In the introduction, indicate links to the works presented by me. What is new in the article in comparison with the studies already performed (M. Wang, Preparation of Ni-diamond composite coating by composite electroplating. Journal of Central South University (Science and Technology) 44(7):2688-2695. 2013. Xiangzhu He, Preparation and Investigation of Ni-Diamond Composite Coatings by Electrodeposition. Nanoscience and Nanotechnology Letters 4(1):48-52, 2012. E.C. Lee, A study on the mechanism of formation of electrocodeposited Ni–diamond coatings, 2001).
4. Due to which the coefficient of friction in nickel-diamond coatings is reduced, provide a scientific justification.
Reviewer 2 Report
Please see the enclosed file for suggestions.

Round 2
Reviewer 1 Report
I received responses to comments.